# Learning Physical Constraints with Neural Projections

**Shuqi Yang**[1*]**, Xingzhe He**[1]**, Bo Zhu**[1]
[1]Dartmouth College, Computer Science Department
*`shuqi.yang.gr@dartmouth.edu`

## Abstract

We propose a new family of neural networks to predict the behaviors of physical systems by learning their underpinning constraints. A neural projection operator lies at the heart of our approach, composed of a lightweight network with an embedded recursive architecture that interactively enforces learned underpinning constraints and predicts the various governed behaviors of different physical systems. Our neural projection operator is motivated by the position-based dynamics model that has been used widely in game and visual effects industries to unify the various fast physics simulators. Our method can automatically and effectively uncover a broad range of constraints from observation point data, such as length, angle, bending, collision, boundary effects, and their arbitrary combinations, without any connectivity priors. We provide a multi-group point representation in conjunction with a configurable network connection mechanism to incorporate prior inputs for processing complex physical systems. We demonstrated the efficacy of our approach by learning a set of challenging physical systems all in a unified and simple fashion including: rigid bodies with complex geometries, ropes with varying length and bending, articulated soft and rigid bodies, and multi-object collisions with complex boundaries.

## 1 Introduction

How does a human being distinguish the motions of a piece of paper and a piece of cloth? A high-school physics teacher might answer that they are both tangentially inextensible but cloth cannot resist any bending force from the normal direction. This raises a further general question for machine perception – what is the most effective representation to characterize a physical system computationally? The answer to this question is fundamental for the design of better neural physics engines [1, 2, 3, 4, 5] to predict the dynamics of various real-world Newtonian systems based on limited observations. In general, a capable neural physics simulator needs to capture the essential features of a dynamic system with a unified computational model, simple network architectures, small training data, and the minimum human inputs for priors. To this end, a vast literature has been devoted into building neural-network models to reason and predict physics. Two of the main areas include to reason the underlying physics by learning local interactions (e.g., the gravitational force between two planets [1]) or by enforcing global energy conservation (e.g., the sum of potential and kinematic energies of a pendulum [6]).

This paper proposes to investigate a third category of approaches to characterize classical physical systems, by establishing neural predictors to learn and enforce underlying physical constraints. The term "physical constraints" broadly defines the various intuitive criteria that the motion of a physical system must satisfy, e.g., a constant length between particles, a fixed angle between two segment pieces, the overlap of joint positions, the non-penetrating geometries for collisions, etc. Such constraints can be either hard or soft, with forms of both equality and inequality. Mathematically, a

set of equality constraints can be expressed as a non-linear equation system $\mathcal{C}(\mathbf{x}) = 0$ with each row $\mathcal{C}_i(\mathbf{x}) = 0$ corresponding to a single constraint exerted on the system. To enforce these constraints over the temporal evolution, a common idea established in the physics simulation communities is to define a projection operator to map the system's current states to a low-dimensional constraint manifold satisfying $\mathcal{C}(\mathbf{x}) = 0$ (e.g., see [7, 8, 9, 10, 11, 12]). By augmenting the dynamics with a Lagrangian multiplier, the projection amounts to the minimization of the following energy form [11]:

$$\min_{\mathbf{x}} g(\mathbf{x}) = \frac{1}{\Delta t^2}(\mathbf{x} - \hat{\mathbf{x}})^T \mathbf{M}(\mathbf{x} - \hat{\mathbf{x}}) + \lambda^T \mathcal{C}(\mathbf{x}), \tag{1}$$

with $\hat{\mathbf{x}}$ and $\mathbf{x}$ as the system's states before and after enforcing the constraints, $\mathbf{M}$ as the mass matrix, and $\lambda$ as a Lagrangian multiplier. The intuition behind Equation 1 is to find the closest point on the constraint manifold to modify the current prediction, e.g., by following the direction of $-\nabla \mathcal{C}$ with a fixed small step in the gradient descent search. The optimization of the energy form in Equation 1 along with its various variations serve as the algorithmic foundation to accommodate a broad spectrum of constraint physics simulators, including articulated rigid bodies [10], collisions [9], contacts [13], inextensible cloth [11], soft bodies [14], and the various position-based dynamics techniques [12, 15, 16, 17], which have recently emerged in gaming industry. Such simulators have also been used to generate datasets for machine learning applications [18, 2]. Meanwhile, the mathematical properties of neural projections have been investigated in the machine learning community (see [19] for examples).

Motivated by the physics intuition behind Equation 1, we devise a new neural physics simulator to unify the prediction of the various dynamic systems by learning their underlying physical constraints. Our main idea is simple: we express the mixed dynamic effects due to all the constraints by one neural network and enforce these constraints by recursively employing the network to correct the system's time-independent states (position). The centerpiece of our learning framework is a neural projection operator that enables the mapping from a current state to a constraint state on the target manifold. The parameters of the operator are trained in an end-to-end fashion by observing the positional states of the system for a certain range of time frames.

Our design philosophy to learn the physics constraints exhibits several inherent computational merits compared with learning relations or energy conservation. *First*, constraints directly relate to human's physical perception. The various physical intuitions, such as length, angle, volume, position, penetration, etc., can be encoded automatically and learned straightforwardly in our neural networks to describe constraints. In contrast, the expression of energy, albeit essential for computational physics, lacks its intuitive counterparts (e.g., many systems do not conserve energy due to their dissipative environments). *Second*, our neural constraint expression describes a system-level relation without requiring any connectivity priors (e.g., it does not need a graph network to specify the relations). This connectivity-free implementation is essential when describing complicated interactions with an uncertain number of primitives. For example, to express the bending effects of a piece of cloth, it requires at least three particles to describe a planar angle in 2D and four particles to describe a bilateral angle in 3D. Such case-by-case priors require expertise in physics simulation and are difficult to obtain beforehand for normal users. *Third*, constraints are a time-independent state variable that can be reasoned with position information only. This alleviates the data requirement to train an expressive neural-network model. Also, the complexity for a neural expression of constraints is low. In our implementation, a small-scale fully-connected network in conjunction with our iterative projection scheme can uncover mixed constraints governing complicated physical interactions. *Last*, from a numerical perspective, enforcing constraints in a numerical simulator essentially amounts to building an implicit time integrator, which is inherently stable and allows for large time steps. This further lowers the training data requirements and enables reliable long-term predictions.

## 2   Related Work

**Neural physics simulators**   Many recent works on learning physics are based on building networks to describe interactions among objects or components (see [20] for a survey). The pioneering works done by Battaglia et al. [1] and Chang et al. [4] predict different particle dynamics such as N-body by learning the pairwise interactions. Following this, the interaction networks are enhanced by a graph network (e.g., [21] [22]) for different applications. Specialized hierarchical representations [18, 2], residual corrections [23], propagation mechanisms [24], linear transitions [25] were employed to reason various physial systems. Besides directly working on particles, there are also many other

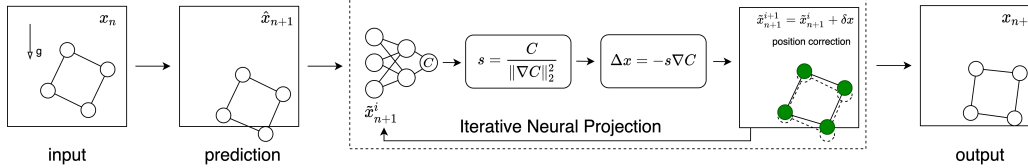

Figure 1: Algorithm overview: Here we show the forward prediction of the motion of a rigid body composed of four particles. In each time step, our neural simulator takes the positions $\mathbf{x}_n$ from time $n$ as input, makes a prediction $\hat{\mathbf{x}}_{n+1}$, employs iterative neural projections to enforce the learned constraints, and outputs the positions $\mathbf{x}_{n+1}$ for time $n+1$.

interaction-based works that make predictions on images and videos [5, 26, 27, 28, 29, 30]. On another front, researchers have also made progress on building learning models that can conserve the important physical quantities, A typical example for energy-conservation is the Hamiltonian system, which was realized by building neural networks to uncover and enforce the Hamiltonian, e.g., see [6], [31], [32], [33], [34], and [35]. Other physical quantities include Lagrangian [36, 37, 38, 39], tensor invariants [40], velocity divergence [39], etc. There exists a broad array of applications for these neural-based simulators, such as controlling legged robots [41] and particle accelerator systems [42, 3]. In addition to neural physics engine, another branch [43, 44, 45, 46] chooses to build differentiable simulators to direct connect physics into learning applications.

**Constraint physics and position-based dynamics**   Simulating constraint physical systems has been thoroughly investigated in computational physics and computer graphics over the last decades. To enforce accurate contact [13], collision [13], articulation [7], and boundary conditions [47], a broad spectrum of methods have been developed to generate visually plausible and physically accurate simulations. Among these approaches, developing fast, stable, but sometimes less accurate, numerical solvers to unify the simulations of the various physical phenomena and provide rich and instant feedback within an immersive virtual environment has received tremendous attention in gaming industry. To this end, position-based dynamics, which enforces a set of manually predefined constraints to correct the particle positions, have achieved great success (see [16] for a survey). Its numerous variations to model different types of physics, such as soft body [12], fluid [15], and unified physics couplings [47], have been contributing to the creation of the vivid physics world in entertainment computing. The nature of the algorithm, namely, to create plausible and fast physics by processing position only, is aligned inherently with the purpose of neural physics simulators that is focused more on perception and control. These algorithms have also been used to design neural networks for data-driven simulation and parameter learning [44].

## 3   Methodology

As in Figure 1, our constraint neural physics simulator works by taking a set of points and predict their future positions. We use $\mathbf{x} \in \mathcal{R}^{2m}$ to denote a vectorized representation of the $m$ positions as $\mathbf{x} = [x_1, y_1, x_2, y_2, ..., x_m, y_m]^T$. For naming conventions, we use the subscript $n$ to denote the time step and the superscript $i$ to denote the projection iteration. Also, we use $\hat{\mathbf{x}}$ to denote the predicted positions and $\tilde{\mathbf{x}}$ for the intermediate projected results. For example, $\hat{\mathbf{x}}_n$ represents the predicted positions for time step $n$, and $\tilde{\mathbf{x}}_n^i$ represents the corrected positions for time step $n$, after $i$ projections.

The goal of the algorithm is to take the point positions from the previous frames and predicts the positions for the future frames. We designed a learning framework as in Figure 1 by processing the point data recursively within a loop embedded with a neural network. The entire framework consists of a training step and a prediction step. The centerpiece of our approach is an iterative neural projection procedure to learn and enforce the various underlying physical constraints (see Section 3). The training step takes a set of frame pairs $(\mathbf{x}_n, \mathbf{x}_{n+1})$ to train the parameters of the neural projector (see Section 4). The prediction step takes the trained parameters to specify the embedded network and forwards the time steps by predicting the positions from $\mathbf{x}_n$ to $\mathbf{x}_{n+1}$ (see Section 5).

## 3.1 Linear prediction

In the prediction step, we calculate the predicted positions $\hat{\mathbf{x}}_{n+1}$ by a linear extrapolation from $\mathbf{x}_{n-1}$ and $\mathbf{x}_n$, i.e., $\hat{\mathbf{x}}_{n+1} = 2\mathbf{x}_n - \mathbf{x}_{n-1}$. This amounts to a linear approximation of the velocity $\mathbf{v}_n$ as $(\mathbf{x}_n - \mathbf{x}_{n-1})/\Delta t$ followed by an explicit Euler time integration $\hat{\mathbf{x}}_{n+1} = \mathbf{x}_n + \mathbf{v}_n \Delta t$. The body force such as gravity is exerted as a prior in the prediction step.

## 3.2 Iterative neural projection

**Network** The mapping from a predicted state $\hat{\mathbf{x}}_{n+1}$ to a constraint state $\mathbf{x}_{n+1}$ is enabled by an iterative neural projection step. The essential component of the projection step is a neural network $\mathcal{C}_{net}(\cdot)$ to learn the mixed hidden constraints by observing its positional states. These constraints range from the constant length (e.g., rod), relative positions (e.g., rigid body), non-penetration (e.g., collision and contact), bending (e.g., cloth), etc. Instead of devising $k$ independent networks to process $k$ constraints, where $k$ could be a prior input, we use a single network to learn the mixed effects of all the constraints employed on the system simultaneously. The input of the network is the values of $\mathbf{x}$ and the output of the network is a single scalar evaluating the satisfaction of all constraints as a whole. In particular, a zero output indicates that all the constraints are satisfied.

**Iterative projection** The network $\mathcal{C}_{net}(\cdot)$ is embedded in an outer loop to recursively enforce the learned constraints on the input $\mathbf{x}$ during the learning process. The iterative projection procedure is motivated by the step of fast projection (SAP) algorithm proposed in [11] and applied in many position-based dynamics simulators [12] to minimize the projection energy given in Equation 1. The three steps to update the positions are shown in Algorithm 1. Specifically, in each iteration $i$, the update for $\tilde{\mathbf{x}}^{i+1}$ for the next iteration is conducted by finding a $\delta\hat{\mathbf{x}}$ based on the current $\tilde{\mathbf{x}}^i$ and $\mathcal{C}_{net}(\tilde{\mathbf{x}}^i)$ to minimize the projection energy in Equation 1. By assuming an identity mass matrix that absorbs $\Delta t$ (implying each particle has an equal contribution), the objective in Equation 1 can be simplified as $g(\tilde{\mathbf{x}}) = \frac{1}{2}\delta\tilde{\mathbf{x}}^T\delta\tilde{\mathbf{x}} + \lambda^T C(\tilde{\mathbf{x}})$, with $\delta\tilde{\mathbf{x}} = \mathbf{x} - \tilde{\mathbf{x}}$ as the error term between the optimized solution $\mathbf{x}$ and the current prediction $\tilde{\mathbf{x}}$.

---

**Algorithm 1:** Iterative Neural Projection

**Input:** Constraint network $\mathcal{C}_{net}(\cdot)$,
predicted positions $\hat{\mathbf{x}}$.

1  $\tilde{\mathbf{x}}^1 = \hat{\mathbf{x}}$ ;
2  **for** $i = 1 \rightarrow N$ **do**
3  $\quad \lambda = C_{net}(\tilde{\mathbf{x}}^i)/|\nabla C_{net}(\tilde{\mathbf{x}}^i)|^2$ ;
4  $\quad \delta\tilde{\mathbf{x}} = -\lambda\nabla C_{net}(\tilde{\mathbf{x}}^i)$ ;
5  $\quad \tilde{\mathbf{x}}^{i+1} = \tilde{\mathbf{x}}^i + \delta\hat{\mathbf{x}}$ ;
6  **end**

**Output:** Projected positions $\mathbf{x} = \tilde{\mathbf{x}}^{i+1}$

---

To obtain the local minimum, we can set the gradient of $g$ to be zero along the search direction to obtain:

$$\frac{\partial g}{\partial \delta\tilde{\mathbf{x}}} = \delta\tilde{\mathbf{x}} + (\nabla\mathcal{C}_{net}(\tilde{\mathbf{x}}))^T\lambda = 0. \tag{2}$$

Also, we employ the Taylor expansion around $\tilde{\mathbf{x}}$ to get

$$\mathcal{C}_{net}(\mathbf{x}) = \mathcal{C}_{net}(\tilde{\mathbf{x}} + \delta\tilde{\mathbf{x}}) \approx \mathcal{C}_{net}(\tilde{\mathbf{x}}) + (\nabla\mathcal{C}_{net}(\tilde{\mathbf{x}}))^T\delta\tilde{\mathbf{x}} = 0 \tag{3}$$

Substituting Equation 3 into Equation 2, we obtain the expression $\lambda = C_{net}(\tilde{\mathbf{x}})/|\nabla C_{net}(\tilde{\mathbf{x}})|^2$, which is the step size coefficient of $\delta\hat{\mathbf{x}}$. The direction of $\delta\hat{\mathbf{x}}$ is the negative gradient of the constraint evaluation calculated by the automatic differentiation on the network.

## 3.3 Constraints

Our neural projection method can handle different types of constraints with a single embedded network, including multiple constraints, inequality constraints, and soft constraints. The examples demonstrating these types of constraints are in Section 5. *First*, our neural projection model can handle multiple constraints automatically by evaluating a mixed objective for all potential constraints underpinning a dynamic system. *Second*, our neural projection method can directly handle inequality constraints automatically without any further modification. The network can detect the situations when an inequality constraint is activated and needs to be enforced. A typical example for learning inequality constraints is to handle collision (see Section 5.1). *Last*, some constraints in physical systems are soft, meaning that they cannot be fully satisfied in an equilibrium state. For example, the bending constraints of an elastic rod or a piece of paper is a type of soft constraints, which means that

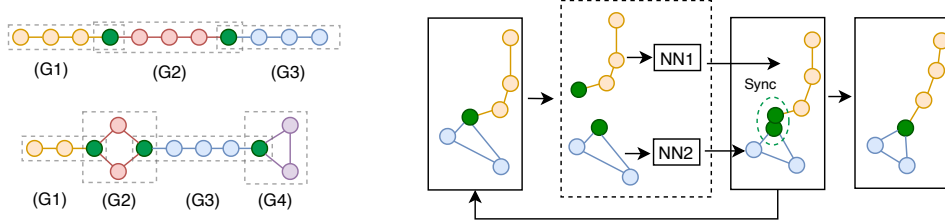

Figure 2: Multi-group point representation (left) and configurable network module connection (right). The points are clustered into a set of overlapping groups as a prior input. The point groups are processed by different network modules and synchronized on the overlapping points (green) in each projection iteration to obtain predictions following all constraints enforced by sub-networks.

it can only be satisfied in a soft way when the system reaches a steady state. To learn a soft constraint, we follow the ideas proposed in [12] by adding a relaxation coefficient in front of $\delta\hat{\mathbf{x}}$ for the position, namely, to update Line 5 in Algorithm 1 as $\tilde{\mathbf{x}}^{i+1} = \tilde{\mathbf{x}}^i + r\delta\tilde{\mathbf{x}}$, with $r$ as the relaxation coefficient.

### 3.4 Hierarchical representations

**Multiple-object representations**   Thanks to the expressive power of our iterative neural projection network, our model can naturally handle multi-object or multi-component systems without requiring any connectivity priors. In our learning model, the null constraint between two points is a kind of knowledge that can be learned automatically by the network. Furthermore, our model can naturally distinguish the strong and weak interactions, such as the stiff and soft pieces in a multi-material body and their articulations, by learning the multiple constraints with the underlying grouping information from the training datasets.

**Multi-group representations**   A single neural projection operator can only process a fixed number of points following a strict order, which could become one of the main weaknesses when learning systems with a larger number of particles. We tackle this problem by employing two strategies: 1) We create a grouping strategy to partition the points into multiple clusters for a hierarchical geometric representation; 2) We implement a configurable mechanism to connect each point group to a pretrained network module to enforce its local constraints. As shown in Figure 2 left, on the data level, we introduce an overlapping grouping representation to partition the point cloud into several groups and specify the shared points across the neighboring groups to propagate the position information. Second, as shown in Figure 2 right, on the network level, we put multiple independent neural networks in parallel within the same projection module to let each of these networks process the position data from one group of points. The connections between the group data and the specific networks can be customized, as long as the dimension of the the input is consistent. The information is synchronized at the end of each projection iteration by averaging the values on the shared points across groups, mimicking the fashion of a classical Jacobi iteration scheme. We want to mention that such parallel processing is not the only way to combine these network components. These small networks can also be combined in a sequence, with the values on the shared points updated directly from the previous group, following the fashion of a multi-color Gauss-Seidel iteration scheme.

---

**Algorithm 2:** Multi-Group Projection

**Input:** NNs $C_{net_1}(\cdot), \cdots, C_{net_M}(\cdot)$,
  Group of positions $\hat{\mathbf{x}}_1, \cdots, \hat{\mathbf{x}}_M$.

1  $\tilde{\mathbf{x}}^1 = \hat{\mathbf{x}}$ ;
2  **for** $i = 1 \rightarrow N$ **do**
3    **for** $j = 1 \rightarrow M$ **do**
4      $\tilde{\mathbf{x}}_j^{i+1} = Project(C_{net_j}, \tilde{\mathbf{x}}_j^i)$ ;
5    **end**
6    Synchronizing $\tilde{\mathbf{x}}^{i+1}$ among groups;
7  **end**

**Output:** Projected positions $\mathbf{x}$

---

**Discussion on connectivity priors**   We want to note that our multi-group representation in conjunction with the customized network modules introduces connectivity priors into our learning model. Such prior input enables our model to learn complex physical systems in a hierarchical way, such as multi-component articulations or a long rod, by dividing the system into multiple pieces that can be handled by a single network. This strategy provides additional learning granularity to restrict the

interaction complexities within a local region by assuming that such interactions can repeat on a higher level. For example, different groups of points can share the same projection network, which enables the customized and modulated design of a learning architecture to fit a specific physical problem. From a relational perspective, such group representation connects our approach to the previous relational networks. In an extreme case, if we group every pair of points in the system, and connect all the groups to the same network, the way of describing the relations in our model will be identical to Interaction Network [1].

## 4   Implementation

**Network architectures** We use a standard architecture to implement our neural projection network. For most of the examples we show in the paper, we use a fully-connected network with $5 \times 256$ elements. For the collision example, the width of the network is changed to $512$. The input of the network is a vector composed of positions of all the points in the system. The output for the network is a single scalar value describing the satisfaction of the constraints. The derivatives $\nabla \mathcal{C}_{net}$ are calculated by auto-differentiation.

**Training data** Our training data covers a rich set of scenarios including rigid bodies, rods, articulations, collisions, contacts, and irregular domains, generated by different types of numerical simulators such as mass-spring, position-based dynamics, and rigid-body solvers. We apply a random force to each particle to perturb the system and physics simulators are used to calculate the position for the next time step. The position of each particle in each time step will be recorded in the training data. We sample the simulation data with the frame rate of 10 fps. For each test, we train our model using data samples ranging from $2048$ to $8102$, with each sample has 20 to 32 frames (see details in Supplementary). In some scenarios (e.g., rigid body), we added noise to several of the simulations to mimic the data collection process from a real-world setting.

## 5   Experiments

We show examples to demonstrate our approach's capability in discovering various constraints and predicting complex dynamics. We implemented our learning model in PyTorch and our physics simulations in C++. We trained all the models using the Adam optimizer [48] on a single Nvidia RTX 2080Ti GPU. We refer readers to the supplementary for detailed parameter settings and animations. We want to highlight that we use a large time step of $\Delta t = 0.1$ second in all our examples.

### 5.1   Examples

**Rigid bodies** Due to the low dimensionality of the constraint space for rigid-body motion (e.g., only three DoFs are allowed in 2D), it is challenging for a neural physics simulator to predict its motion purely based on positional information. We showcase the capability of our model in accurately predicting rigid-body motions with only the position input. As illustrated in Figure 3, we tested two rigid bodies, one composed of 4 and the other composed of 16 particles. In each example, we added a velocity change with the same magnitude but opposite direction on two particles to rotate the body. The results (see Figure 9 and 10) showed that our model can simultaneously preserve the rigid shape and predict accurate long-term dynamics as compared to the ground-truth.

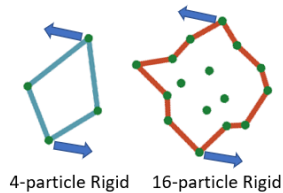

4-particle Rigid      16-particle Rigid

Figure 3: Rigid body rotation

**Rope with mixed constraints and group representations** As shown in Figure 4, we test our model by predicting the motion of a 2D rope. We showcase the ability of our method to enforce both stretching and bending constraints by learning from simulation data. We demonstrate that our model can effectively learn the mixed effects of these two constraints without any prior input. We also show the feature of our group-representation by learning the dynamics of a piece of short rod (with 8 particles) and predict the motions of a long rod composed of three short pieces (with each two pieces share

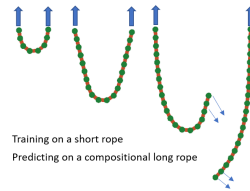

Training on a short rope
Predicting on a compositional long rope

Figure 4: Rope with bending

two points). Further, we show our model can predict motions that it never observed in the training set, e.g., the falling of rod after releasing one of the two fixed points, by naturally enforcing the neural constraints during the prediction steps.

**Articulated objects with multi-object interactions** In this example, we demonstrate that our model can learn constraints for multiple components within a single network. We set up our simulation as an articulated rigid body connected by a soft rod. The learning model is expected to discover three types of constraints affecting the dynamics in a mixed fashion: 1) the rod length and bending; 2) the rigid body position and orientation; 3) the articulation between the rod and the body on a single point. As shown in Figure 5, we show that our neural projection model can precisely predict the articulated

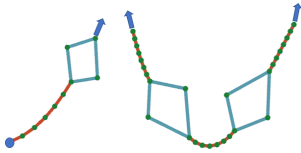

Figure 5: Articulated body

motion satisfying all the underlying constraints, by observing the point-cloud data with position information only. We further demonstrate our model's efficacy in predicting complex systems by grouping multiple components to create a chain of articulated bodies. As shown in Figure 10 and the video, our model can correctly enforce the learned physical constraints in the new scenario.

**Collisions and contacts with unknown environment** In this example, we demonstrate the capability of our model in learning complex, hierarchical interactions within an unknown environment. In the simulation, we set up two rigid-bodied squares interacting with each other by collision and contact. The rigid bodies are also interacting with a spherical terrain. In the simulation, each rigid body is represented by 16 points. To set the tasks to be more challenging, we build the training dataset by sampling the positions of 4 points (the same 4 points are used in all

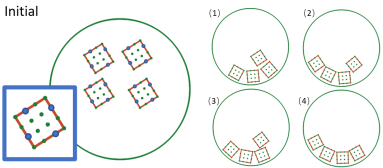

Figure 6: Collision and contact. The blue dots mark the four sample points.

samples) that are not the four corners of the square to decouple the observation points and the contact points. We expect our learning model to reason the full set of physics rules governing this miniature world, consisting of the concepts of multiple object, the constraints of rigid bodies, collisions, contacts, and their interactions with an unknown environment. Our model can effectively uncover all the underlying constraints along with their interactions by accurately predicting the dynamics of the two rigid bodies on all stages. In particular, the model can automatically learn the contacts between the bodies and the unknown environment. To further demonstrate the predicative capability of our model, we extend the learned model to predict the motion of two groups of objects, i.e., with 4 rigid squares interacting within the same environment. The results in Figure 6 demonstrate that our model can successfully predict the complex interactions among four bodies satisfying all the underlying constraints that the model has never seen in the training dataset.

**Three-dimensional examples** We show that our neural projection model can work for 3D physics predictions. As shown in Figure 7, we predict the motion of a 3D rigid body (top) and a piece of inextensible cloth (bottom). In particular, we learn the cloth's bending and stretching constraints. The cloth dynamics is modeled by 64 points in 3D space and we subdivide its geometry into $7 \times 7$ groups, with each group composed of $2 \times 2$ particles.

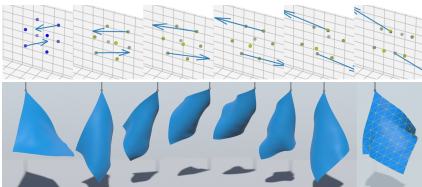

Figure 7: 3D rigid body and cloth.

## 5.2 Comparison to other approaches

For all four examples, we made comparisons between our iterative neural projection method and the groundtruth, a naive MLP network to predict the correct term for the next timestep, and our implementation of the Interaction Net [1]. We also inject noise into the training data to mimic the real-world data sampling process. For each comparison, we plotted two error measurements: the mean squared error of positions compared to the groundtruth trajectories (Figure 9) and the error of the underlying constraints (Figure 10). It is noteworthy that the error statistics we showed in Figure 10 is from a single prediction example. We further demonstrate the averaged errors of all test cases, each with 200 examples, in Table 1. The animations are shown in Figure 8. As illustrated in the

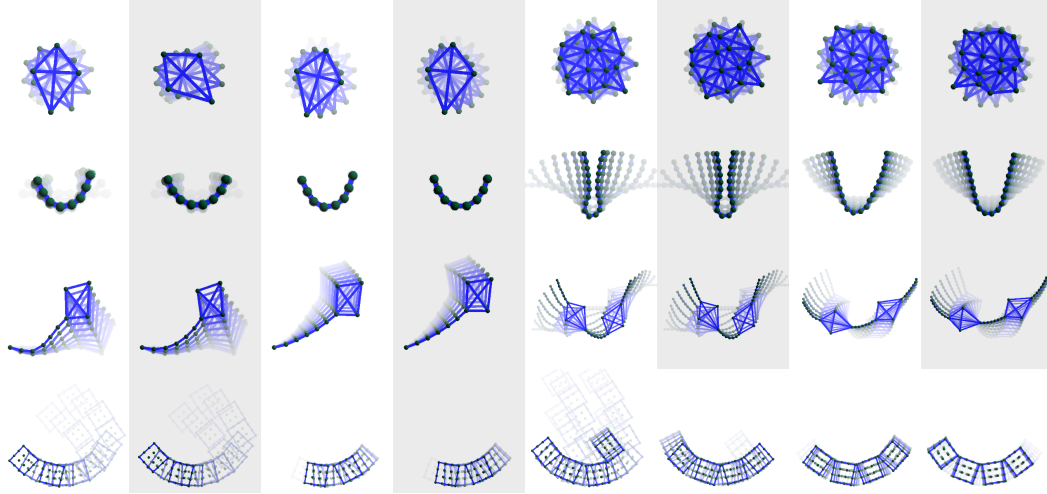

Figure 8: The predicted dynamics (white background) and groundtruth (grey background). Motion blur was used to animate the motion trajectory. For the last example, our naive position-based simulator fails on this multi-object case, therefore we have no groundtruth and only show our results.

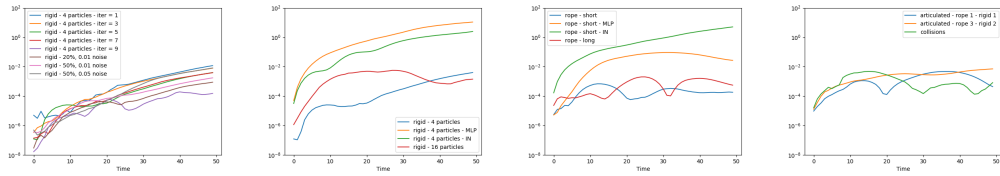

Figure 9: The MSE of the positions compared to the groundtruth. From the left to right: rigid body, rope, articulated body, and collision.

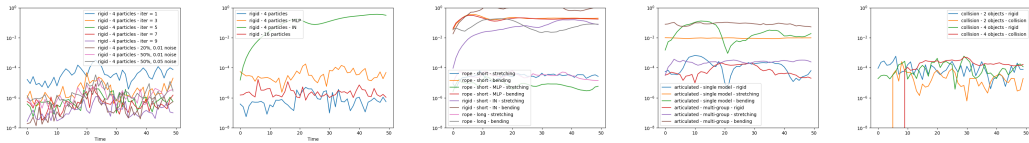

Figure 10: The constraint satisfaction in different examples. The measured constraints include 1) and 2) rigid body: distance between two arbitrary points on the body; 3) rope: distance between two neighboring points and the bending angle between two incident segments on the rope; 4) articulated: the distance constraint on the rigid body part and the stretching and bending constraints on the rope part; 5) collision and contact: the distance between corners on a rigid body and the distance between a corner of a rigid body and the terrain. The numbers are calculated as the mean value of all constraints.

plots, our iterative neural projection models enable trajectory errors on the level between $10^{-4}$ and $10^{-3}$, while a naive implementation and an IN model produces errors around $10^{-2}$. For the various constraint errors, our model reliably produces a precise satisfaction of all underlying constraints around $10^{-4}$, while other models produce models two orders of magnitude higher in most cases.

To showcase the effects of $\Delta t$ on explicit methods such as IN, we trained another two interaction network models using smaller time steps ($\Delta t = 0.01s$ and $\Delta t = 0.001s$) and more regularly sampled data to show that the interaction-based approaches are sensitive to the time-step size. The two models generate average constraint errors of $5e-5$ (for $\Delta t = 0.001s$) and $0.026$ (for $\Delta t = 0.01$) on 200 test samples. In contrast, our approach obtained the constraint error of $5.3e-5$ with $\Delta t = 0.1s$.

Table 1: Average constraint satisfactions of 200 samples * 50 frames/sample predicted simulation results. Ri-1 and Ri-2 are two rigid bodies examples; Ro-1 and Ro-2 are two rope examples; Ar-1 and Ar-2 are two articulated examples; Co-1 and Co-2 are two collision examples.

|  | Ri-1 | Ri-2 | Ro-1 | Ro-2 | Ar-1 | Ar-2 | Co-1 | Co-2 |
|---|---|---|---|---|---|---|---|---|
| Shape | 4.7e-7 | 2.3e-6 | – | – | 4.15e-5 | 3.5e-4 | 2.7e-5 | 3.2e-5 |
| Stretch | – | – | 5.3e-5 | 6.9e-5 | 4.0e-3 | 1.0e-3 | – | – |
| Bend | – | – | 2.2e-1 | 7.2e-2 | 1.5e-1 | 4.7e-2 | – | – |
| Collision | – | – | – | – | – | – | 2.4e-5 | 1.1e-4 |

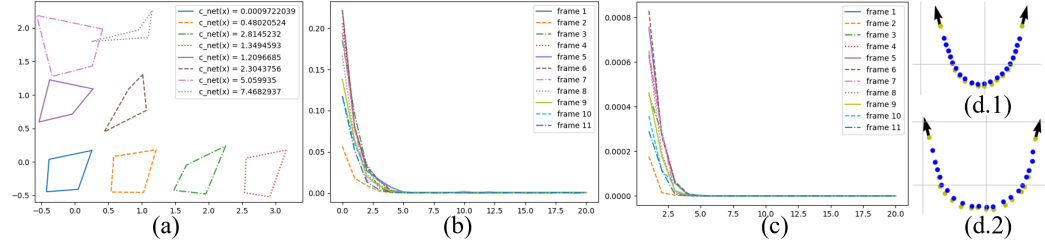

Figure 11: Learned Constraints. (a): $C_{net}$ values given differnet inputs; (b): $C_{net}$ values of each iterations of 10 frames; (c): $\delta \mathbf{x}$ of each iterations of 10 frames; (d): Relaxing the constraints for a simple rope (d.1), we can get a softer and more elastic rope(d.2).

## 5.3  The learned constraints

We show the physical correctness of our learned constraint values by comparing them with the analytical ones in three aspects. First, in Figure 11a, we show the consistency between the constraint satisfaction (the shape of a rigid body) and the magnitude of the learned constraint error ($C_{net}$). Next, as in Figure 11b and c, we plot the learned constraint $C_{net}$ (network output) and the positional correction $\Delta \mathbf{x}$ against iteration steps. We observed that both quantities' convergence after 5 iterations (the same iteration number in training). In Figure 11d, we further show that the learned constraints can replace the original ones used in our rope simulator. In particular, when we relax the learned constraint to different extent, we can get different elastic behaviors as if we tune the constraint values in the original simulator.

## 6  Conclusion

We devised a new approach for neural physics simulations by learning their underpinning constraints. Our method consists of an iterative neural projection procedure to discover the physical constraints from training data and enforce these constraints for accurate forward prediction. Compared with previous interaction-based or energy-based methods, we showed by various examples that our constraint-based neural physics engine is versatile, intuitive, robust, and fast, with the ability to learn complex physics rules governing challenging scenarios.

**Limitations and Future Work** There are several limitations of our approach. First, the effects of soft constraints are currently controlled by a stiffness as a hyper-parameter. Incorporating it in the network architecture will further automate the learning process for soft bodies. Second, the multiple networks that uncover component-wise constraints are trained in separate models. Our next step will involve an end-to-end design to discover all constraints on a global system level. Third, our learning model is currently focused on Lagrangian models, i.e., by assuming all the data points are Lagrangian particles that interact in a material space. Another broad array of physical systems driven by the Eulerian models, such as fluid, are currently out of the radar of our neural projection method. We plan to further investigate the Eulerian nature underpinning dynamics projection to support the learning of this category of systems. Such extension will require more advanced network architecture such as graph networks to accommodate the effective learning of the invariant constraints based on locally varying interactions. On another front, we plan to devise more neural network algorithms that are motivated by fast physical simulation techniques to bridge the communities between machine learning, scientific computing, and visual computing.

## Broader Impact

This research constitutes a technical advance by employing constraint projection operations to enhance the prediction capability of physical systems with unknown dynamics. It opens up new possibilities to effectively and intuitively represent complicated physical systems from direct and limited observation. This research blend the borders among the communities of machine learning and fast physics simulations in computer graphics and gaming industry. Our model does not necessarily bring about any significant ethical considerations.

## Acknowledgement

We acknowledge the anonymous NeurIPS reviewers for their insightful feedback. This project is support in part by Dartmouth Neukom Institute CompX Faculty Grant, Burke Research Initiation Award, and NSF MRI 1919647.

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
