[Supplementary Material 1 · appendix.pdf]

# Supplementary: Learning Physical Constraints with Neural Projections

## A  Animated Video

We refer the readers to the supplementary video for the animated results of all examples.

## B  Additional Implementation Details

### B.1  Training Data

We provide all the information for the training dataset in Table 1 and Figure 1. The time step used for all the training examples is $dt = 0.1s$.

Table 1: Training Data Details

| Model | #Samples | #Frames per Sample |
|---|---|---|
| Rigid-1 | 2048 | 20 |
| Rigid-2 | 8192 | 20 |
| Rope | 4096 | 32 |
| Articulated | 6000 | 32 |
| Collisions | 8192 | 32 |

### B.2  Network Architectures and Training Details

All the models use LeakyReLU as the activation function and use fully-connected layers as the basic units. They are trained using ADAM optimizer, with different training parameters as shown in Table 2.

Table 2: Network Architectures and Training Details

| Model | Architecture | Batch size | Init_lr | lr_step | lr_gamma | Epoch | Iter |
|---|---|---|---|---|---|---|---|
| Rigid-1 | $[8, 256, 256, 256, 256, 1]$ | 256 | 1e-3 | 20 | 0.8 | 600 | 5 |
| Rigid-2 | $[8, 256, 256, 256, 256, 1]$ | 512 | 1e-3 | 20 | 0.8 | 1000 | 8 |
| Rope | $[8, 256, 256, 256, 256, 1]$ | 256 | 1e-3 | 20 | 0.8 | 1000 | 10 |
| Articulated | $[8, 256, 256, 256, 256, 1]$ | 512 | 1e-3 | 20 | 0.8 | 1000 | 8 |
| Collisions | $[8, 512, 512, 512, 512, 1]$ | 256 | 1e-3 | 20 | 0.8 | 1000 | 10 |

### B.3  Algorithm Complexity

Let $N$ denote the number of parameters of the neural network, and $Iter$ denote the number of iterations of the projection. The time and space complexity of the inferring stage of our method are both $O(N * Iter)$. Since the gradient of the network is used in forward process, the network's second-order gradient needs to be calculated in training stage. This makes the training complexity to be $O(N^2 * Iter)$.

# C   Additional Visualization for Training and Prediction

Here we provide additional visualizations as in Figure 1 and Figure 2 to demonstrate the training data and predicted process. We refer the readers to the video for more detailed illustrations.

Figure 1: We used a dataset composed of different types of physics simulations on points. Each simulation was run with some randomized parameters for a short time period. The simulations include four-point rigid body (top left), short rod (top right), collision and contact (bottom left) and articulation (bottom right). All the simulations (and the predictions) took large time steps to showcase the stability of our algorithm.

Figure 2: The visualization of the predicted points (yellow) and the neural projected points (blue) for different examples, including rigid (top left), articulated body (top right), collision between two bodies (bottom left) and collision between multiple bodies (bottom right). The red dots visualize the additional points used in generating simulation data.



[Supplementary Material 2]

# Supplementary: Learning Physical Constraints with Neural Projections

## A  Animated Video

We refer the readers to the supplementary video for the animated results of all examples.

## B  Additional Implementation Details

### B.1  Training Data

We provide all the information for the training dataset in Table 1 and Figure 1. The time step used for all the training examples is $dt = 0.1s$.

Table 1: Training Data Details

| Model | #Samples | #Frames per Sample |
|---|---|---|
| Rigid-1 | 2048 | 20 |
| Rigid-2 | 8192 | 20 |
| Rope | 4096 | 32 |
| Articulated | 6000 | 32 |
| Collisions | 8192 | 32 |

### B.2  Network Architectures and Training Details

All the models use LeakyReLU as the activation function and use fully-connected layers as the basic units. They are trained using ADAM optimizer, with different training parameters as shown in Table 2.

Table 2: Network Architectures and Training Details

| Model | Architecture | Batch size | Init_lr | lr_step | lr_gamma | Epoch | Iter |
|---|---|---|---|---|---|---|---|
| Rigid-1 | $[8, 256, 256, 256, 256, 1]$ | 256 | 1e-3 | 20 | 0.8 | 600 | 5 |
| Rigid-2 | $[8, 256, 256, 256, 256, 1]$ | 512 | 1e-3 | 20 | 0.8 | 1000 | 8 |
| Rope | $[8, 256, 256, 256, 256, 1]$ | 256 | 1e-3 | 20 | 0.8 | 1000 | 10 |
| Articulated | $[8, 256, 256, 256, 256, 1]$ | 512 | 1e-3 | 20 | 0.8 | 1000 | 8 |
| Collisions | $[8, 512, 512, 512, 512, 1]$ | 256 | 1e-3 | 20 | 0.8 | 1000 | 10 |

 # C   Additional Visualization for Training and Prediction

 Here we provide additional visualizations as in Figure 1 and Figure 2 to demonstrate the training data
 and predicted process. We refer the readers to the video for more detailed illustrations.

Figure 1: We used a dataset composed of different types of physics simulations on points. Each simulation was run with some randomized parameters for a short time period. The simulations include four-point rigid body (top left), short rod (top right), collision and contact (bottom left) and articulation (bottom right). All the simulations (and the predictions) took large time steps to showcase the stability of our algorithm.

Figure 2: The visualization of the predicted points (yellow) and the neural projected points (blue) for different examples, including rigid (top left), articulated body (top right), collision between two bodies (bottom left) and collision between multiple bodies (bottom right). The red dots visualize the additional points used in generating simulation data.