[Reviews · NeurIPS 2020]

Review 1

Summary and Contributions: The authors propose a model with an inductive bias for constraint satisfaction via a constrain satisfaction network C that outputs a single constraint satisfaction scalar given the state of a system. The inference step works as follows: 1. Update the position using linear extrapolation. 2. Iteratively correct the updated position to optimize for constraint satisfaction using C and its derivatives to maximize constraint satisfaction while keeping the correction small. 3. After a few correction iterations (sometimes with multi-group iterations), output the final corrected position. The most interesting aspect is that rather than having to supervise the constraint satisfaction network directly, the network is implicitly end-to-end via next step prediction. I think this is the main contribution of the paper: a new inductive bias inspired on position based dynamics and constraint satisfaction. EDIT: I thank the authors for their elaborate responses, as well as the other reviewers. In line of that I would like to keep my acceptance recommendation at 6.

Strengths: The approach is creative and works as intended. To me it was not immediately obvious that it would be possible to learn a constraint satisfaction function implicitly end-to-end via this procedure. The paper shows the potential of a new inductive bias, which could be useful in many applications.

Weaknesses: From the results, it seems that separate models are trained for each of the systems, but within each system, there is not that much procedural generation in the structure/relative distances of the system particles, just on the initial positions/velocities. Do you expect this to work in cases where there is more procedural variation in the relative positioning of the points (e.g. if sometimes the rigid it is a square, sometimes an arbitrary trapezoid). I guess this could work if you added another input to C with the previous state, or with some reference distances, but it does not seem it would work on the current form of the model, since it would be impossible for the C function to tell whether the constraints are being satisfied or not, just by looking at the position of the points, since it does not know if the constraints to be satisfied should be that of a square or that of a specific trapezoid. Similarly, I wonder how much of the context of the other particles the model is relying on to infer how systems should collide against a wall. For example, if we were making predictions for a system with a single particle that in a single timestep would bounce elastically of a wall, I wonder if the system would always put the particle right at the wall (the linear prediction would move it past the wall, and the constraint satisfaction would put it back right at the wall, where the constraint is satisfied with minimal displacement, but would not make it bounce back off the wall. Then in the next step the linear extrapolation, would essentially do the same thing once more, and then beyond that the particle would become permanently stuck at the wall once two consecutive positions place it at the wall). I think because of this, it may be better to give the previous positions, and the current position to the constraint satisfaction model (or a sequence of recent positions if you wish to make sure the inputs are is Markov). While the key idea of the paper is really good, it would be good to see a maybe more matured version of the model, that is able to handle a more general set of problems. More generally, because most of the systems described here are quite simple, and the generalization modes quite limited, it is a bit unclear if this would actually work (or how expensive the iterations would be) for much larger real-world systems. Otherwise the generality of the current technique is unclear. I am also slightly surprised by how weak the IN baseline looks, could you give more information about the IN? Is this a single IN, or a stack of INs? First I am surprised to see that it is weaker than the MLP baselines, since It has been shown multiple times that Deep INs/GraphNetworks make better predictions than MLPs for particle systems, although maybe this does not happen here due to the lack of procedural variation of the structures, which makes the MLP perform reasonably well. Also, other work such as [21] specifically show systems that require constraint satisfaction when predicting the 3d position and orientation of each of the bodies forming the control systems, and in that case the two layer GraphNetwork is able to satisfy the constraints very well, keeping the systems together, and certainly much better qualitatively that what the baseline video for the IN shows in this paper. More recent work on Graph Networks (https://proceedings.icml.cc/static/paper_files/icml/2020/6892-Paper.pdf) has shown even more complex constraint satisfaction on solids and fluids in containers, being handled by the INs I think the right use of Interaction Networks / Graph Networks here should not be as a baseline transition model, but as different function approximator for the Constraint Network, specifically, a single Graph Network (or a stack of them) would be able to summarize a graph as a single scalar in a compositional way (Similar to how Graph Networks are integrated with Hamiltonian Networks, which also output a scalar, in this work https://arxiv.org/abs/1909.12790). If using a Graph Network as the constraint function for these particle systems, you should be able to both train, and test on systems with a variable number of nodes. This might let you possibly remove the multigroup projection part of the paper, which seems the less elegant, and possible the less general purpose part of the model. I would be happy to raise my score given more additional evidence on those directions.

Correctness: The methodology seems correct. Perhaps the Broader impact claim that "This research constitutes a major technological breakthrough", is overclaiming a bit, unless this is shown to be successful in some real world application.

Clarity: The paper is clear and easy to follow.

Relation to Prior Work: The relation to prior work seems adequate. I would like to maybe see more analogies with the Hamiltonian Neural Networks work, as they resemble a lot in spirit (network that outputs a scalar that is optimized end to end via assumptions about what the network and its derivatives should represent), and maybe see more the paper use the inductive bias terminology when talking about the contributions.

Reproducibility: Yes

Additional Feedback: Since the technique learns the constraint function end to end, technically the model just provides an inductive bias inspired by constraint satisfaction, but it would be possible that the network is training C(x) for a slightly different purpose than intended. This is similar to how Hamiltonian Neural Networks are trained, in that the Hamiltonian is never directly supervised, but it is possible to later inspect if the function has actually learned a Hamiltonian. It would be nice to see some experiments investigating the scalar outputs of the C(x) function, and verifying that it is very close to 0 for states that satisfy the constraint, and how it increases for states that don’t. My intuition is that the C(x) function would be part constraint satisfaction, and part transition model? (line 183) “The information is synchronized at the end of each projection iteration by averaging the values on the shared points across groups, mimicking the fashion of a classical Jacobi iteration scheme” What if this process breaks important constraints? Is this why the outer loop in Algorithm 2 is needed? Equation (2) seems to be designed so delta x (the correction) is also minimal at each iteration. However, because the process is repeated in a loop, and delta x is redefined at each iteration, potentially the total accumulated delta x could accumulate/grow quite a bit. Does the process usually converge to delta_x = 0. Does the process generalize well if you use more/less iterations at test time, compare to train time? Algorithm 1: “Output: Projected positions x” but bold(x) is not defined, I assume bold(x) = x^tilde^n+1 Algorithm 2: The “Project” function does not seem to be defined anywhere, is this a single projection step, or does it include an additional internal loop, such as in algorithm 1? (line 267) “To set the tasks to be more challenging, we build the training dataset by sampling the positions of 4 points that are not the four corners of the square to decouple the observation points and the contact points.” Are the same 4 points sampled for all examples in the training dataset or does each example have a different set of points. The actual loss does not seem to be mentioned anywhere in the main paper or supplementary material. Even if this is just MSE between predicted (after constraint satisfaction iterations) and the next step state, it would probably be good to add a sentence. The training is done with one-step data, but never learned through sequence right? This means that during training C only sees as inputs states with a minimal deviation from the constraint satisfaction, but potentially, at test time, when a full rollout trajectory is calculated, the deviations from the constraint satisfaction could be larger, and reach regimes that C never saw during training, which would cause weird exploding dynamics. Did you ever observe this?


Review 2

Summary and Contributions: The paper proposes to implement the projection step in position-based dynamics (PBD) for physics simulation through neural networks. In PBD, positional constraints such as collisions, bending, stretching are resolved iteratively on groups of particles. The approach uses a neural network to predict a function indicating constraint violation with a scalar value and resolves the constraints through gradient descent and a Jacobi iteration like scheme on particle groups. Few qualitative examples are reported and an unclear quantitative evaluation is made with comparison to interaction networks [1].

Strengths: * The paper proposes a straightforward extension to PBD to learn the accumulated constraint function using neural networks and to integrate it with the constraint projection loop of PBD. The method is demonstrated in scenarios involving rigid and soft bodies, compositions of rigid and soft parts and collisions in small particle systems. * An evaluation is made on several scenarios and a comparison is made with interaction networks.

Weaknesses: * The particle systems considered in the experiments are rather small. What are the limits in system size and what are the limiting factors? * The paper provides insufficient details of the training/test data. What is the size of the benchmark problems, what are the physical parameter/initial state variations in the dataset? * The paper should provide a systematic quantitative benchmarking for size or structure of the estimation problems. * Which scenarios are generated with the PBD baseline simulation, and doesnt this give an advantage for the proposed method over interaction networks? * The network is trained on the specific physical parameters of the objects such as mass, stiffness, elasticity. It does not generalize well to new scenarios with different physical properties which limits the generality and applicability of the method. * No comparison is made to state of the art methods such as - Hu et al. ChainQueen: A Real-Time Differentiable Physical Simulator for Soft Robotics. ICRA 2019. * The paper uses single examples in Figs. 3-6 and claims that they show that predictions are "accurate" or "precise" without justification. A thorough quantitative evaluation and analysis on a signficiantly large dataset with various parametrizations and initial states should be made, and the accuracy of the method should be assessed. * The paper mentions a network which uncovers componentwise constraints but it is unclear what this should be and how it works. From l.173 it seems the grouping mechanism is a hand-crafted algorithm. The paper should provide further algorithm details how the grouping is determined and how this affects performance.

Correctness: The evaluation does not support claims on the accuracy/preciseness of the method.

Clarity: The paper is well written and clear.

Relation to Prior Work: The paper overlooks important work on particle-based physics engines such as - Hu et al. ChainQueen: A Real-Time Differentiable Physical Simulator for Soft Robotics. ICRA 2019.

Reproducibility: No

Additional Feedback: == UPDATE (post-rebuttal) == The author response has addressed several of my concerns. Experiments should be improved by providing further statistical analysis and 3D results. The MSE measures should be grounded through relation with the object size or object motion to improve their interpretability.


Review 3

Summary and Contributions: This paper attempts to learn a neural physics simulator. However, instead of learning a "forward prediction model", as commonly done, they employ a linear extrapolation model to predict future state, followed by a "neural projection" operator that 'corrects' the state of the simulation so as to obey physical constraints. The neural projection operator is evaluated in multiple scenarios that involve learning dynamics of rigid/non-rigid objects, and also collisions and articulated interactions among multiple objects.

Strengths: [S1] This paper takes a contrary approach to typical "forward model" learning approaches, and attempts to learn a physics simulator by using a "neural projection" operator/network. Building off of work in the real-time interactive graphics community, they use the popular projection operator (essentially a neural network that outputs a scalar), to predict whether or not a system currently satisfies constraints. The gradient of the projection operator then specifies a descent direction that "stabilizes" (pushes towards constratint satisfaction) the system. Overall, this idea is very interesting, and experimental evaluations across a wide range of challenging scenarios makes this paper very appealing. [S2] Most approaches that learn neural simulators eg. [1, 2, 6] assume knowledge of the structure of the system (eg. number of particles, etc.). This work relaxes the assumption and discovers structure (as a constraint) [S3] The authors make an interesting connection between the multi-group representations and interaction networks (although I'd have liked to see some empirical results on this end; the current explanations lend a hand-wavy flair). [S4] The supplementary video attachment helps appreciate the proposed approach better. I've also looked at the code whenever in doubt, and that helped me navigate the paper better.

Weaknesses: [W1] One aspect of the paper that can certainly be improved is the experimental evaluation. For one, the authors only treat interaction networks as a baseline for most experiments. It would be interesting to see how the current approach fares against problems setup in the interaction networks paper (particle systems, force fields, etc.). [W2] Arguably, the experimental scenarios (while challenging from a neural networks standpoint), seem to indicate some scalability challenges for the technique. For instance, Some of the most complicated collision cases show 4 cubes with 4 points each, colliding with a sphere. It would be interesting to see how the approach fares with more points, more objects, and non-convex shapes. [W3] One other aspect that goes undiscussed is the gradient of the projection operator. While the authors indicate that this is computed through an automatic differentiation framework, gradient computations are typically expensive (the authors use pytorch, where this is aggravated). It would be helpful to see more analysis in this direction, and will be good to know if there are any smart 'tricks' employed herein. [W4] Another limitation of the current manuscript is that most of the experiments involve 2D objects, and complete observability is assumed. It would be beneficial to add a discussion of some of these limitations in a revision.

Correctness: To the best of my knowledge and abilities, the method, evaluation protocols are correct. Wherever I've found the empirical methodology lacking, I've outlined it as a corresponding "weakness".

Clarity: The paper is extremely well-written. The central arguments are well-structured, and convincingly put forth. I thoroughly enjoyed reading this paper!

Relation to Prior Work: The related work section was commensurate with what's expected of a conference submission. I'd have liked to see some discussion of old-school projection neural network literature. See [R1] for an example.

Reproducibility: Yes

Additional Feedback: Overall, I enjoyed reading this paper, and I think this is a refreshingly novel take on neural physics simulation. However, in its current form, it comes forth more of a "proof-of-concept". Another worrisome aspect of the paper is lack of evaluation with other approaches. In particular, it would be essential to compare against approaches that assume complete knowledge of the system. A discussion of compute / timing requirements could be a good addition to the paper/supplementary material. I'd have liked better for the code to have a 1-1 correspondence with the terminology and notation of the paper (more of a nitpick). [R1] A Projection Neural Network and Its Application to Constrained Optimization Problems. IEEE Transactions on circuits and systems. 2002. UPDATE: Upon reading the other reviews and author response, I find most of my concerns adequately addressed. I think the most novel/interesting aspect of this paper is that it discovers structure in physical simulations, and does this in challenging scenarios too. While I think this work could have presented more experiments/comparisons, I also feel the paper in its current form is well-poised to stimulate future work in this area. My revised score reflects this change of mind about the paper. However, I strongly encourage the authors to add a section detailing the drawbacks of the proposed approach. For one, they never clarified my concerns on the runtime of each backward pass (it appears to be significant). Further, other assumptions (such as perfectly elastic collisions, etc.) haven't been clearly stated in the paper. Incorporating these aspects will greatly strengthen the paper.


Review 4

Summary and Contributions: This paper proposes to use neural networks in order to predict the behaviors of physical systems. The key idea is using neural networks to approximate the function of complex physical constraints and performing projection with the neural networks. The authors also propose hierarchical representations to handle complex objects. Experiments show that the proposed method can generalize to different object configurations.

Strengths: - The proposed method is quite general. It is not based on any strong assumptions of the objects/scenes and the experiments show that the method is able to simulate different physical systems with various physics property/configuration. - The proposed method is novel and highly related to the NeurIPS community. It can inspire future research on combining deep learning with physics simulation.

Weaknesses: - The authors only conduct experiments on 2D toy examples with a small number of particals. Since most physics simulation methods simulate the real-world objects/scenes in 3D space, I strongly recommend the authors to demonstrate the simulation results of complex 3D objects such as clothes/hairs. - The evaluation is not sufficient. I think the authors should conduct evaluations on the proposed hierarchical representations on complex scenes to see how the grouping method affects the simulation accuracy. ========================================================= The authors clearly addressed my major concerns about the scalability. After reading the response and other reviewers' comments, I would like to keep my original rating and recommend acceptance. However, I strongly recommend the authors to present more experimental results on complex objects/scenes in the paper and also add a section detailing the drawbacks of the proposed approach as suggested by R3.

Correctness: Yes

Clarity: Yes

Relation to Prior Work: Yes

Reproducibility: Yes

Additional Feedback: None

[Author Response · NeurIPS 2020]

We thank all the reviewers for their constructive suggestions and insightful comments. We will address all the suggested expositional changes such as the algorithm complexity, details on training, and adding all the suggested references. We will also release our codebase with clear comments and naming conventions to ensure reproducibility.

**Role of $C$ and Quantitative Evaulations (R1,R2,R3,R4)** The network output $C$ corresponds to pure constraint satisfaction. No transition component was involved in the final output, owing to the projection nature of our iterative algorithm. In this sense, the constraint satisfaction functioned as the key criterion for our evalua-

tions to measure the effectiveness of the learning model (e.g., see Fig 9). This criterion was fundamentally different from the previous work and we believed it to be a vital point to quantitatively reveal the nature of a constraint dynamic system. This fact is further evidenced by our latest experiment shown in the inset figures. We plotted the learned constraint $C_{net}$ (network output), the real, observed constraint $C_{real}$ (analytical expression), and the magnitude of the positional correction $\Delta\mathbf{x}$ against iteration steps. We observed that all three quantities converge to zero after 5 iterations (the same iteration number we set in training). Further, we calculated the Pearson correlation coefficient of our learned $C$ and real $C$, using 1000 random frames, each with 5 iterations, and got a statistically significant correlation of $0.914$ (for linear relations). This observation indicates a clear physical meaning of the network $C$ that is almost identical to its analytical counterpart. To show this, we further operate on the learned $C$ by relaxing its value and observe different levels of constraint satisfaction. For instance, we can predict rope / rigid behaviors with different stiffness by relaxing the network $C$ to different extents (see figure (d)(e) above).

**Comparison with IN and Other Approaches (R1,R2,R3)** We chose to make direct comparisons with IN because we believe it is the family of approaches most relevant to ours, which aims to uncover *unknown* dynamics from limited observation. We did not compare our method with differentiable physics solvers, such as ChainQueen, which assume *known* governing equations. One of the main reasons that our model can outperform IN is due to its implicit nature, realized by time-independent correction, which is inherently suited for tackling stiff systems such as rigid and articulated bodies. Such systems are challenging for explicit transitional methods due to the timestep restriction (imagine the difficulty on simulating a rigid body using springs with infinite stiffness). Currently, we reported the timestep size ($\Delta t = .1$ for all examples) in Supplementary. We will highlight this in Results and add further discussions endorsed by a new experiment we conducted to demonstrate the different timestep sensitivities between the two models. We are also happy to incorporate comparisons with other models, but to the best of our knowledge, our projection paradigm is the only approach that can uncover constraints in such a simple and end-to-end fashion.

**Real-World 3D Applications (R1,R2,R3,R4)** There was no technical barrier that prevents our approach from being used in predicting 3D physical systems. Here we show a 3D cloth example, as asked by R4, to showcase its capability in predicting more complicated 3D physics. We are happy to extend all our four examples to 3D to better demonstrate its scalability. On the other hand, we also want to argue that the main difficulty on reasoning a real-world physical system lies in the system's range of stiffness rather than its number of DoFs. E.g., a rigid body has 6 effective DoFs only, yet its dynamics is challenging to obtain using a transitional learning model which does not take a rotational prior and has the same parameter size as ours (0.3M).

The four examples we showed in our manuscript covered dynamic systems exhibiting a broad range of stiffness and different types of constraints, which we believe can characterize the main portion of real-world solid systems (rigid, soft, articulation, and collision). Last, as mentioned in Limitation, we acknowledge that our algorithm can process solids only (rigid, soft, or any system with a fixed material space). This model cannot predict Eulerian systems with temporally varying local relations (e.g., fluid). R1 made insightful suggestions on tackling such challenges by incorporating GNNs into neural projection. We will discuss this direction in Future Work.

**R1 Individual Comments** *Collision*: All the collisions in the dataset are currently inelastic; *Alg 2*: Yes, the outer loop is for averaging corrections among groups and the projection function is the same as Alg 1.; *Does $\Delta x$ converge to zero?* Yes! *More/fewer iterations at test time?* More is fine (because of projection) but fewer does not work; *Are sample points the same in Fig 6?*: Yes; *Error accumulation*: The constraint errors do not accumulates but the trajectory errors do; *IN and MLP*: We used a single layer IN for comparison. MLP outperforms IN because it predicts correction only.

**R2 Individual Comments** *Size of benchmark*: See Supplementary B1; *Parameter variation*: We randomized initial conditions for position, orientation, and external forces, all ranging from $[-5, 5]$; *Problem structure*: We used the analytical expressions to measure position, length, and angle constraints; *Simulation*: Our model is not sensitive to simulation algorithms as far as the underlying constraints can be observed from data; *Single example*: The plots for Fig 3-6 were used specifically to accommodate the animated examples. We had obtained and will incorporate statistical data with more parameter variations; *Grouping*: Yes, the grouping information was set as a prior input.

**R3 Individual Comments** *Gradient of projection*: The gradient was calculated using the standard auto-differentiation.

[Meta-Review · NeurIPS 2020]

The paper proposes an approach for modeling dynamical systems. Instead of directly modeling the transition function it learns a linear forward mapping together with a constraint satisfaction function that can be used as a learned neural projection operator. The reviewers agree that this an interesting, non-obvious, and novel idea that could stimulate considerably future work. The reviewers also raised several concerns in particular with respect to the quality of the evaluation which in the form presented in the submission is not entirely satisfying (scope of the experiments; quality of the baselines). Overall, however, the reviewers were satisfied with the author response and recommend acceptance. We strongly encourage the authors to take into account the reviewers’ comments and improve the empirical evaluation and to be more upfront about the limitations of the proposed approach in the final version.